# Assessment of muscle oxygenation following eccentric exercise-induced muscle damage using near-infrared spectroscopy

Chris McManus[1]*, Kelly Murray[1], Elizabeth Welbourn[2], Julie Double[2], Henry Chung[1], Sally Waterworth[1], Ben Jones[1], Chris Cooper[1]

1 School of Sport, Rehabilitation and Exercise Sciences, University of Essex, Colchester, United Kingdom,
2 School of Life Sciences, University of Essex, Colchester, United Kingdom

* cmcman@essex.ac.uk

## Abstract

Exercise-induced muscle damage (EIMD) disrupts muscle structure and function, impairing recovery and performance. Near-infrared spectroscopy (NIRS) offers a non-invasive method to assess muscle oxygenation, yet its sensitivity to EIMD-related changes under practical conditions remains unclear. This study examined whether resting tissue saturation index (TSI) and sprint-phase oxygenation kinetics are altered following EIMD. Seventeen recreationally active males were assigned to a control (n = 5) or experimental group (n = 12). The experimental group performed 10 × 10 eccentric squats at 80% 1RM. Resting TSI, sprint-phase desaturation/resaturation, countermovement jump (CMJ), wellness, and creatine kinase (CK) were measured at baseline and 1, 24, 48, 72, and 96 hours post-exercise. Control data provided typical error and smallest worthwhile change thresholds for reliability and interpretation. No statistically significant differences were found over time or between groups for CK, performance, or NIRS variables (P > 0.05). However, effect size-based inferences revealed small to extremely large effect sizes in CMJ, wellness, CK, and key NIRS metrics. Notably, desaturation amplitude and rate during Sprint 1 showed large effects at 1 h post-EIMD, while resting TSI demonstrated a small increase at 24 h. These observations indicate that resting TSI and selected sprint-phase NIRS indices exhibit small but systematic changes in temporal association with an eccentric squat protocol that induces EIMD, and suggest that NIRS may have the potential to contribute to non-invasive characterisation of muscle status in this context.

## Introduction

Exercise-induced muscle damage (EIMD) is a common consequence of unaccustomed eccentric exercise and poses a significant barrier to athletic performance and recovery. Clinically, EIMD is characterised by delayed onset muscle soreness

**Data availability statement:** All data underlying the findings of this study are publicly available via the Open Science Framework (OSF): McManus, C. (2025, July 8). Assessment of Muscle Oxygenation Following Eccentric Exercise-Induced Muscle Damage Using Near Infrared Spectroscopy. Retrieved from https://osf.io/uq3kp. In addition, the custom MATLAB script used for data extraction and digitisation is included as Supplementary Data (S1) with this manuscript.

**Funding:** This study was supported by the Departmental Research Fund (DRF) in the School of Life Sciences at the University of Essex. The funders had no role in study design, data collection and analysis, decision to publish, or preparation of the manuscript.

**Competing interests:** The authors have declared that no competing interests exist.

(DOMS), strength loss, reduced range of motion, and localised swelling. These symptoms typically emerge 24–72 hours post-exercise and reflect structural and inflammatory changes [1,2]. At the cellular level, eccentric contractions disrupt sarcomeres, cause Z-line streaming, and increase sarcolemma permeability. These alterations trigger calcium-dependent proteolysis and inflammatory cascades that elevate circulating markers such as creatine kinase (CK) and interleukin-6 (IL-6) [3–5]. These changes impair force production, alter motor unit recruitment, and delay recovery, compromising performance in both endurance and high-intensity activities [6,7]. Consequently, there is a critical need for non-invasive, real-time monitoring tools to assess muscle damage and inform recovery strategies.

Near-infrared spectroscopy (NIRS) has emerged as a promising non-invasive technique for evaluating the physiological effects of EIMD. By measuring relative concentrations of oxygenated ($O_2$Hb) and deoxygenated haemoglobin (HHb) in muscle tissue, NIRS provides insights into local oxygen delivery, extraction, and utilisation [8–10]. However, the utility of NIRS in EIMD contexts remains debated. Evidence indicates that eccentric squats can slow deoxygenation ([HHb]) kinetics during subsequent cycling, suggesting impaired oxygen extraction efficiency [11]. In a different EIMD model, exhaustive downhill walking has been shown to accelerate oxygen desaturation during isometric contractions at 30%, 50%, and 80% MVC, with desaturation and resaturation kinetics remaining altered for up to four days post-exercise [12]. These findings, despite their differing directional effects, suggest that muscle damage can disturb normal deoxygenation kinetics and may reflect microvascular dysfunction and increased energy demands for muscle repair [12]. In contrast, other studies have reported no change in oxyhaemoglobin desaturation/resaturation kinetics following eccentric cycling, suggesting that oxidative capacity may be preserved despite the presence of DOMS [13]. Moreover, a study employing eccentric knee-extensor exercise to elicit EIMD found that tissue saturation index (TSI%) and HHb at rest and during incremental exercise remained unchanged 48 hours post-EIMD, despite a notable decline in force production [14]. This highlights how muscle-level oxygenation can be preserved even when functional capacity is compromised. These conflicting findings may arise from methodological differences in exercise modality (e.g., squats vs. cycling), measurement conditions (resting occlusion vs. dynamic exercise), analytical processes (correcting for blood volume) and muscle groups studied (vastus lateralis vs. gastrocnemius) [15,16].

Unaccustomed or high-intensity eccentric contractions are particularly disruptive because the mechanical strain on a reduced number of active sarcomeres can lead to focal microtears, oedema and elevated intramuscular pressures [17,18]. These effects are amplified in protocols involving high-force decelerations (e.g., downhill running) or repeated lengthening contractions (e.g., resistance training), which impair local blood flow and oxygen diffusion [9,19]. While such protocols reliably induce EIMD, their impact on resting muscle oxygenation and dynamic oxygen kinetics during high-intensity efforts is not well understood. For instance, one study found no changes in tissue saturation index (TSI%) 48 h after electrically stimulated eccentric knee extensions, despite significant reductions in critical torque [20]. This suggests

that task-specific metabolic demands (e.g., supramaximal sprints vs. submaximal isometric efforts) may modulate NIRS sensitivity to EIMD.

Current NIRS protocols often rely on vascular occlusion tests or prolonged exercise tasks, which can limit their applicability in applied sport settings [12,13]. For example, estimates of muscle oxidative capacity are commonly derived from protocols that combine NIRS with repeated arterial occlusions using pneumatic cuffs inflated to suprasystolic pressures, with multiple inflations and deflations and careful control of blood volume artefacts required to obtain valid estimates [16]. In addition, depending on the size and position of the occlusion cuff, these procedures can be uncomfortable or painful, particularly when cuffs are placed proximally on the thigh [21]. Such requirements for specialised occlusion systems, trained operators and participant tolerance reduce the feasibility of repeated occlusion-based testing for routine monitoring in team-sport or field environments. In contrast, brief maximal or repeated sprint tests on a cycle ergometer are already embedded within athlete testing and monitoring, have demonstrated reliability and ecological validity, and can be performed using equipment that is commonly available in high-performance settings [22,23]. However, it is not yet known whether NIRS indices obtained during such short, high-intensity efforts are sufficiently sensitive to detect changes in muscle oxygenation following EIMD.

This study has two primary aims: (1) to determine whether resting TSI in the vastus lateralis is altered following a standardised eccentric squat protocol, and (2) to assess the sensitivity of NIRS-derived variables, including desaturation rate during a sprint and resaturation kinetics during recovery, to EIMD-related changes in muscle oxygenation. By evaluating a protocol that combines resting TSI with a short series of maximal cycle sprints, we intend to test a NIRS approach that can be integrated into existing performance testing without the need for vascular occlusion or prolonged submaximal exercise. Addressing these aims will help to establish the viability of NIRS as a practical tool for monitoring EIMD in applied settings and inform the design of recovery monitoring strategies that are feasible outside of highly controlled laboratory environments.

## Materials and methods

### Participants

Seventeen healthy, recreationally active young men (mean ± SD; age 22 ± 5 years; body mass 76.9 ± 9.4 kg; stature 1.74 ± 0.26 m) were prospectively recruited for this study between 25 April 2014 and 25 May 2014. All participants were non-sedentary and reported regular engagement in sport or exercise at a social, non-elite level (e.g., university club basketball, football or volleyball, or gym-based resistance and/or aerobic training) on average at least twice per week in the months preceding the study. All were free of injury and pre-existing medical conditions, were non-smokers, were not using prescription, anti-inflammatory or analgesic medication and had not been exposed to altitude in the two weeks prior to participation. Participants were instructed to avoid strenuous exercise for 72 hours before baseline testing and to refrain from additional structured exercise or sports participation for the duration of the study, and none had taken part in similar EIMD research protocols in the preceding six months. Ethical approval was obtained from the University of Essex Research Ethics Committee. Written informed consent was provided by all participants prior to inclusion in the study, in accordance with the Declaration of Helsinki.

### Experimental procedures

Participants were allocated into either an experimental (EXP) or a control (CON) group. Allocation was sequential rather than randomised. The first five participants were allocated to the CON group to determine the day-to-day variation of the dependent variables. This group did not undergo the EIMD protocol. A priori power analysis was conducted to determine the minimum sample size required to assess within-subject reliability across six repeated measures in a control condition. Targeting a conservative threshold for moderate reliability (ICC = 0.50) for the primary NIRS outcomes (resting TSI and sprint phase TSI indices), it was determined that a sample size of five participants

would provide >99% power to detect a true ICC of 0.50 ($\alpha = 0.05$) [24]. In commonly used interpretative guidelines, ICC values below 0.50 are considered to indicate poor reliability, whereas values between 0.50 and 0.75 reflect moderate reliability [25]. We therefore selected 0.50 as a minimal threshold to establish that day-to-day variation in the control condition was at least in the moderate range, acknowledging that higher ICC values would be preferable for applied decision-making.

The subsequent twelve participants were allocated to the EXP group to complete the full protocol. A priori power analysis was conducted based on previously published %$O_2$ saturation data following EIMD [12]. Data were digitised using a custom MATLAB script (See S1 File) based on pixel-coordinate calibration of figure axes [12], (Fig 4a). Values from five time points (baseline and Days 2–5 post-exercise) were used to estimate within-subject variability and absolute change in tissue oxygenation. This yielded a large effect size (Cohen's F = 1.39), for which a repeated-measures ANOVA indicated that a sample size of 12 participants would provide 80% power to detect a significant main effect of time ($\alpha = 0.05$). A target power of 80% is widely recommended as a minimum standard for hypothesis-driven studies, balancing the risk of Type II error against the practical constraints of intensive physiological protocols [26]. In addition, the resulting sample size is consistent with previous NIRS studies examining muscle oxygenation responses to eccentric exercise and EIMD, which have typically included approximately 9–13 participants [11–13].

All participants reported to the laboratory for testing on six separate occasions within eight days. All experimental sessions were completed in a controlled laboratory environment (temperature: $18 \pm 1°C$; relative humidity: $65 \pm 5\%$). On the first visit, participants completed a maximal incremental cycle test, followed by a 30-minute seated rest and then a 1-repetition maximum (1RM) barbell squat test. After these maximal assessments, participants were familiarised with the remaining procedures and equipment: under investigator supervision, they were shown the NIRS set-up and anthropometric measurements, coached on countermovement jump technique, and performed a small number of practice efforts (approximately three CMJ and three 6-second sprints on the cycle ergometer) with ample rest and verbal feedback. This sequencing was chosen so that any minor fatigue associated with familiarisation did not carry over into the incremental or 1RM tests performed on that day. Dependent variables were measured pre-EIMD, 1 hour post-EIMD, and 1–4 days post-EIMD (Table 1). The CON group followed the same schedule, except they did not complete the EIMD protocol. For all post-EIMD visits, a NIRS device was positioned on the vastus lateralis while participants rested supine for 10 minutes. Subsequently, participants completed a series of assessments in the following order: baseline NIRS reading, venous blood sample for CK, mid-thigh girth, wellness questionnaire, countermovement jump (CMJ) test, and repeated cycle sprint test.

**Table 1. Overview of experimental procedures throughout preliminary visits and post-EIMD exercise.**

| Measure | Pre-1 (Day 1) | Pre-2 (Day 4) | EIMD Exercise | Post 1 h (Day 4) | Post 24 h (Day 5) | Post 48 h (Day 6) | Post 72 h (Day 7) | Post 96 h (Day 8) |
|---|---|---|---|---|---|---|---|---|
| Incremental cycle test | ✓ | | 10 x 10 eccentric barbell squats (80% 1RM) (EXP group only) | | | | | |
| 1RM barbell squat | ✓ | | | | | | | |
| Resting NIRS | | ✓ | | ✓ | ✓ | ✓ | ✓ | ✓ |
| CK | | ✓ | | ✓ | ✓ | ✓ | ✓ | ✓ |
| Mid-thigh girth | | ✓ | | ✓ | ✓ | ✓ | ✓ | ✓ |
| Wellness | | ✓ | | ✓ | ✓ | ✓ | ✓ | ✓ |
| CMJ | | ✓ | | ✓ | ✓ | ✓ | ✓ | ✓ |
| Repeated cycle sprint | | ✓ | | ✓ | ✓ | ✓ | ✓ | ✓ |

Abbreviations: RM, repetition maximum; CK, creatine kinase; NIRS, near-infrared spectroscopy; CMJ, countermovement jump; EIMD, exercise-induced muscle damage; EXP, experimental group [EIMD group].

## Baseline assessments

**Maximal incremental cycle test.** A bicycle ergometer (Lode, Excalibur Sport, Netherlands) was adjusted for comfort. Participants performed a 5-minute 50W warm-up at $80 \pm 10$ rpm. During the brief pause before the incremental test, they were permitted to perform light, self-selected stretching if desired, but this was not prescribed and was intended solely to maximise comfort rather than to influence test performance. The incremental test began with 2 minutes at 1 W·kg$^{-1}$ followed by increases of 0.5 W·kg$^{-1}$ every 2 minutes. Participants were required to maintain a cadence of $80 \pm 10$ rpm throughout. The test was terminated when cadence fell <70 rpm for >5 s or upon volitional exhaustion. $VO_{2max}$ was verified if three criteria were met: a $VO_2$ plateau ($VO_2$ increases <2.1 mL·min·kg$^{-1}$), RPE of 19–20 on Borg's 15-point scale, a respiratory exchange ratio of >1.1, or heart rate >90% of age-predicted maximum (220-age) [27]. Maximal aerobic power (MAP) was calculated as the power output of the last fully completed stage plus the proportion of the final, incomplete stage completed (time in final stage ÷ full stage duration) multiplied by the power increment between stages.

Respiratory gases were continuously monitored. Participants wore a mask connected to an impeller turbine assembly (Hans Rudolph, Kansas, USA). Gas concentrations were sampled using electrochemical ($O_2$) and infrared ($CO_2$) analysers (Vyaire CPX, Mettawa, Illinois, USA). Before each test, gas analysers were calibrated using known gas concentrations and ambient air. $VO_{2max}$ was defined as the highest 30-s time-averaged $VO_2$ value [28], calculated from breath-by-breath data exported in 5-s intervals.

**1RM Barbell Squat.** A Smith machine (Rogue Fitness, UK) was used to assess 1RM. Participants warmed up with 5–10 reps, rested 3 minutes between sets, and progressed via load increments of 10–20 kg until near-maximal single repetition. Failed lifts prompted a 5–10 kg reduction for re-attempt of 1RM. A standard 20 kg barbell with weight plates was used. Two researchers ensured correct technique and depth for consistency.

## Exercise-induced muscle damage protocol

Participants performed 10 × 10 eccentric-only barbell squats using a Smith machine at 80% of concentric 1RM. Each repetition began from standing (knee extended at 180°), followed by controlled lowering over 3 seconds to 90° knee flexion. Researchers then assisted the return of the barbell to the start position to avoid concentric loading. Sets were separated by 2 minutes of rest. Due to participants' unfamiliarity with the eccentric-only movements, consistent verbal and visual cues were provided to reinforce proper technique. This specific emphasis on the eccentric phase has been shown to impact subsequent muscle function [29].

## Dependant variables

**Creatine kinase.** Plasma CK concentration was assessed from venous blood drawn from the antecubital vein into 4 ml heparinised Vacutainer® tubes. Whole blood samples were centrifuged at 3500 rpm for 10 minutes at 4°C, after which the plasma supernatant was carefully aspirated into clean microcentrifuge tubes. Plasma samples were stored at −80°C until batch analysis and thawed at ambient temperature prior to assay. A 500µl aliquot of each sample was placed into a 10KDa cut-off spin concentrator and centrifuged at 4,000 rpm for 20 minutes to exclude small molecules, such as ATP and NAD. Subsequent CK activity in the plasma was determined using the Ab155901 colorimetric assay kit (BioVision, UK), following the standard Abcam protocol.

**Anthropometry.** Mid-thigh girth was measured using an inextensible Rosscraft measuring tape (Rosscraft Innovations, Vancouver, Canada) with a precision of 1 mm in accordance with protocols set by the International Society of the Advancement of Kinanthropometry (ISAK) [30], and all measurements were executed by an ISAK-accredited anthropometrist. Body mass was determined using a digital scale (Tanita SC-330S, Amsterdam, The Netherlands).

**Wellness score.** Participants' wellness was assessed using a composite metric comprising five self-reported variables: sleep quality, fatigue, soreness, stress, and mood [31]. Data were collected at every visit, following resting NIRS and anthropometric assessments and prior to undertaking any physical activity. A 5-point Likert scale was used to facilitate

accurate and consistent responses, supplemented by brief qualitative descriptors. For instance, sleep quality was rated from "1" (very poor) to "5" (excellent). This "higher is better" approach was applied across all five domains. A cumulative wellness score was then calculated by summing the scores of the five variables, giving a maximum possible score of 25. This brief questionnaire was used as a global indicator of perceived wellness and readiness to train, rather than as a domain-specific validated measure of delayed-onset muscle soreness.

**Countermovement jump.** Participants executed three CMJs, each interspersed with a 2-minute rest, on a contact-timing jump mat (SmartSpeed, Fusion Sport, Queensland, Australia) interfaced with a handheld pocket computer (iPAQ, Hewlett-Packard, Palo Alto, CA, USA). Initiating from an upright stance, participants descended to a self-chosen depth before immediately propelling upward, aiming for maximal elevation. The SmartSpeed software estimated CMJ height (m) based on flight time [32]. The maximum height achieved across the three trials was used for analysis.

**Repeated cycle sprint.** Ten minutes after the countermovement jump assessment, participants were positioned on the cycle ergometer, using the same set-up as for the incremental ramp test. A 5-minute warm-up was conducted at a 50W resistance.

The main protocol comprised ten 6-second maximal effort seated sprints, each separated by a 24-second passive recovery period. During recovery, participants held the pedal cranks in a horizontal position, remained seated, upright and arms fully extended. The ergometer was set in cadence-dependent (linear) mode. The 'linear factor', which governs the flywheel braking resistance as a function of power output/cadence$^2$, in which power output is determined by:

$$\text{Power (W)} = L \times cadence^2 \tag{1}$$

Where cadence is expressed in revolutions per minute (rev·min$^{-1}$) and L is the individual linear factor that sets flywheel resistance. For each participant, the linear factor was individualised using the maximal aerobic power (MAP) obtained during the preceding ramp VO$_2$max test. A target sprint intensity equal to 300% of MAP at a reference cadence of 120 rev·min$^{-1}$ was specified, and the linear factor was calculated as:

$$L = \frac{3 \times \text{MAP}}{120^2} \tag{2}$$

With this configuration, if a participant attained 120 rev·min$^{-1}$ during a sprint, the resulting power output would correspond to approximately 300% of their MAP, with higher or lower cadences producing proportionally higher or lower power outputs according to the cadence squared relation. The choice of 120 rev·min$^{-1}$ as the reference cadence is consistent with evidence that peak cycling power typically occurs at crank velocities around 120–160 rev·min$^{-1}$ [33] and with previous intermittent sprint protocols that set resistance so that supramaximal work rates are achieved at 120 rev·min$^{-1}$ [34]. The 300% MAP target was selected pragmatically, informed by pilot testing and by prior work that has used values of approximately 220–270% of ramp test peak power for repeated 6–30 second sprints on the same ergometer [34].

Pedal cadence and workload (instantaneous mechanical power of the brake) were exported from the ergometer's analogue outputs to a PowerLab 8/30 system (ADInstruments, Oxford, UK) and sampled at 10 Hz using LabChart 5 Pro software. For each 6-second sprint, mean power was calculated as the average of all 10 Hz power samples within the sprint window, and peak power was defined as the highest 10 Hz power value attained during that sprint. Sprint 1 mean and peak power were retained as single-sprint descriptors of performance in each session. In addition, repeated sprint performance across the 10-sprint set was characterised using summary metrics derived from all sprints. Mechanical work for each sprint was calculated as the time integral of power:

$$\text{Work (kJ)} = \frac{\text{mean power (W)} \times 6s^{-1}}{1000} \tag{3}$$

 

Total work (kJ) was calculated as the sum of sprint work across sprints 1–10. To quantify performance decrement across the set, a percentage decrement index was calculated from mean power as the difference between the idealised total (best sprint mean power × number of completed sprints) and the observed total (sum of mean power across sprints), expressed relative to the idealised total [35]. The same decrement approach was also applied to peak power values to provide a complementary index of repeated sprint fatigue.

**Near Infrared Spectroscopy (NIRS).** A portable NIRS device (PortaMon; Artinis Medical Systems BV, The Netherlands) is a compact (83 × 52 × 20 mm) and lightweight (84 g) system. The PortaMon is a dual wavelength (760 and 850 nm), continuous wave system, containing three pairs of LEDs with a source-detector spacing of 30, 35, and 40 mm. The PortaMon uses the modified Beer-Lambert law to calculate the changes in absolute concentration of oxy(+myo) haemoglobin ($O_2Hb$), deoxyhaemo(+myo)globin (HHb), and total hemo(+myo)globin (tHb) and spatially resolved spectroscopy (SRS) methods to measure the tissue saturation index. The tissue saturation index (TSI) was expressed in % and calculated as $[O_2Hb]/([O_2Hb] + [HHb])$ x 100. TSI (%) is independent of NIR photon path length in muscle tissue and was calculated using the SRS method [36]. Data were collected at a sampling rate of 10 Hz.

Tissue Saturation Index (TSI) was used for analyses instead of HHb, $O_2Hb$ and tHb. This decision was informed by evidence suggesting that TSI offers a more robust indicator of muscle oxygenation, especially under conditions where blood flow is variable. This includes during exercise due to the muscle pump effect [37].

The NIRS device was securely positioned on the vastus lateralis' belly, equidistant between the greater trochanter and the lateral epicondyle of the femur [38]. A surgical marker pen was used to mark the position of the NIRS device in order to identify any movement during testing and ensure the NIRS device was placed at the same location on subsequent days [39]. Mid-thigh skinfold at the NIRS optode site was measured in triplicate using Harpenden skinfold callipers, and the median value was recorded. Adipose tissue thickness was approximated as half the skinfold value. The NIRS probe used a source–detector separation of 35 mm, which corresponds to an effective sampling depth of ~17–18 mm in muscle. Waterproof adhesive tape was used to anchor the device, preventing signal contamination from external light sources and any relative movement between the optodes, detector, and the participant's skin. To further ensure optimal adhesion, any body hair near the sensor placement area was removed, and participants were instructed to avoid moisturising that specific region on the test day. The same researcher attached each device and applied the adhesive tape for all testing procedures.

**NIRS data processing.** Resting NIRS measurements were recorded after a 10-minute supine rest period, during which participants were instructed to remain still. Data were collected over a 1-minute period, while the investigator visually observed the live NIRS data to ensure signal stability (TSI variation <2%). Mean 1-minute TSI% was used for resting TSI analyses.

Although participants performed ten repeated sprints during each testing session, NIRS analysis was restricted to three time points: during the first sprint, the immediate recovery period following that sprint, and the recovery period following the 10th sprint (Fig 1). This approach aligned with our aim of evaluating NIRS as a practical, field-compatible tool, thereby avoiding more extensive analyses of multiple sprints or prolonged submaximal/occlusive protocols. By focusing on these specific time points, we were able to obtain a distinct desaturation and resaturation profile while reducing the likelihood of noise introduced by participant repositioning, movement artefact, or inconsistencies such as false starts. This strategy also minimised the accumulation of artefactual data often associated with longer or more complex protocols, supporting a reliable and practical assessment of muscle oxygenation dynamics across visits.

NIRS-derived variables in sprint cycling exercise are subject to rapid and large perturbations. These can be observed as large oscillations in the signal response due to the mechanical effects of muscle contraction on local blood flow, and therefore, smoothing of NIRS signals is often implemented [40]. Therefore, to enhance the quality of the NIRS signals and minimise high-frequency noise, a digital filtering approach was employed. Specifically, the raw 10 Hz NIRS data were subjected to a 10th-order, zero-phase shift, low-pass Butterworth filter using the '*filtfilt*'

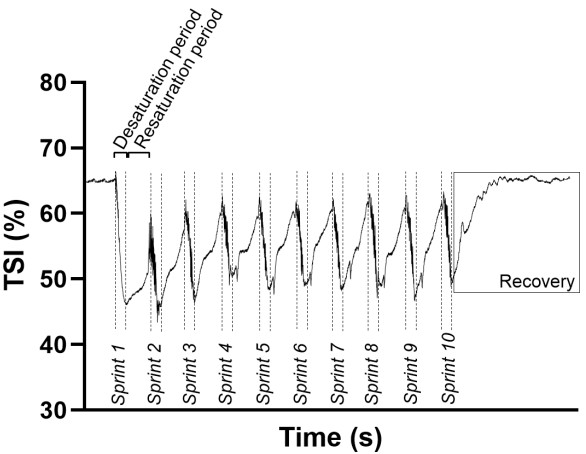

**Fig 1. An illustrative example of the repeated sprint protocol and NIRS-derived TSI, indicating sprint and recovery phases.**

package in Python (Python, 3.8). The Butterworth filter, renowned for its flat frequency response in the passband and sharp cutoff, was chosen to provide a balance between preserving the integrity of the physiological signal and eliminating unwanted high-frequency artefacts [40]. The cutoff frequency for the filter was set at 0.1 Hz. This decision was informed by previous research, which demonstrated that this cutoff frequency effectively isolates the hemodynamic fluctuations of interest from potential high-frequency noise sources such as motion artefacts and instrument noise [40]. The zero-phase shift characteristic of the filter was achieved by forward and reverse filtering, ensuring that the output signal was not shifted in time relative to the input, preserving the temporal alignment of the physiological events. After filtering, the resultant NIRS data were exported to Microsoft Excel for further analysis. Signal peaks and nadirs were identified using standardised algorithms, representing key physiological transitions during sprint and recovery phases. Rather than applying fixed time windows to define sprint and recovery periods (e.g., 6s sprint, 24s recovery), these phases were delineated based on the inflection points in the oxygenation signal itself, allowing for a more physiologically responsive and data-driven approach to analysis. This has been described as a 'rolling' approach and is recommended over the use of predetermined time windows to minimise underestimation of NIRS-derived parameters [40].

Rate of desaturation (sprint) and resaturation (recovery) (%/s) was calculated as;

$$\text{Rate } (\%/\text{s}) = \frac{\Delta \text{TSI}\%}{s}$$

(4)

With $\Delta TSI$ indicating the absolute tissue saturation change and $s$ representing the time of the phase.

The duration between peak and nadir values was used to determine time. NIRS data relating to the repeated sprint protocol includes Sprint 1 absolute TSI change ($\Delta$TSI%), rate of change (%/s) and time between peak and nadir (s) for sprint and recovery time points. TSI was also analysed following the final sprint (Sprint 10) to characterise recovery kinetics. Recovery of TSI after Sprint 10 was modelled in GraphPad Prism (version 10.0.2, Boston, Massachusetts, USA) using a one-phase association monoexponential function:

$$Y(t) = Y_0 + (A - Y_0) * \left(1 - \exp^{(-k*t)}\right)$$

(5)

Where $t$ is time in seconds from the end of Sprint 10, $Y_0$ is the TSI value at the end of the sprint ($t = 0$), $A$ is the asymptotic recovery value, and $k$ is the rate constant. Data were fitted from the end-sprint value ($t = 0\,s$) to 120 s of passive recovery. From each individual fit, the model-derived half-recovery time ($t_{1/2}$, s) was calculated as:

$$t_{1/2} = \frac{\ln(2)}{k}$$

(6)

representing the time required for TSI to recover 50% of the change between $Y_0$ and $A$. This half-recovery time was used as the primary kinetic outcome for reliability analysis and to examine the effect of EIMD on recovery of muscle oxygenation.

## Statistical analysis

Data are presented as either means ± SD or means with 90% confidence intervals (90% CI), as specified. Typical error (TE) was initially calculated for NIRS, performance, perceptual and blood variables in the control group, providing a site-specific threshold for detecting meaningful change [41]. TE was derived by computing the standard deviation of individual difference scores between sessions, divided by √2. Reliability was assessed over five trials using Hopkins' spreadsheet method [42]. The smallest worthwhile change (SWC) was defined as 0.2 times the between-participant standard deviation. To assess test-retest reliability, the Intraclass Correlation Coefficient (ICC) with a 95% confidence interval was calculated using a two-way mixed model, assuming absolute agreement for average measurements [25]. ICC values were interpreted as: < 0.5 = poor, 0.5–0.75 = moderate, 0.75–0.9 = good, and >0.90 = excellent [25]. In line with commonly used guidelines, an ICC of 0.50 was considered the minimal threshold at which reliability ceases to be regarded as poor, which informed the a priori sample size calculation for the control group. Both the lower and upper bounds of the 95% CI are reported [25]. Additionally, the coefficient of variation (CV) was used to quantify individual variability, calculated as the ratio of the standard deviation to the mean for each participant (expressed as %), then averaged across the group to provide a measure of overall variability relative to individual mean performances. CV thresholds were defined as: < 10% (excellent), 10–20% (good), 20–30% (moderate), and ≥30% (poor) [43], and are used here as descriptive guides to aid interpretation rather than as strict cut-off criteria.

Baseline measures for both CON and EXP groups were assessed for normality using the Shapiro-Wilk test and for homogeneity of variances via Levene's test. Group differences at baseline were subsequently analysed using either an independent samples t-test or, in cases of variance in homogeneity, the Welch's t-test. Repeated measures ANOVA was used to determine group x time interaction for all dependent variables. Mauchly's test of sphericity was employed to verify the assumption of equal variances of the differences between all combinations of related groups. Where the assumption of sphericity was violated, the Greenhouse-Geisser correction was applied to adjust the degrees of freedom accordingly. Where significant interactions were observed, Bonferroni post hoc tests were conducted. All analyses were performed using SPSS (version 28, IBM), with a statistical significance set at α = 0.05.

To evaluate the impact of the EIMD intervention, mean change values were derived by subtracting baseline from post-intervention measurements, with 90%CI calculated for each value. To ensure only meaningful effects were considered, the larger of the TE or SWC (calculated from control participants) was used as the threshold for significance [41]. Changes were deemed "unclear" if the mean change aligned with the typical error or if the 90% CI overlapped zero [41]. In cases where the mean change exceeded the TE or SWC and the 90% CI excluded zero, standardised effect sizes were reported [44]. Interpretation of these effect sizes adhered to Hopkins' established thresholds: Small (0.2), Moderate (0.6), Large (1.2), Very Large (2.0), and Extremely Large (4.0) [44].

## Results

Table 2 presents the reliability statistics for the NIRS parameters collected from the control participants (n = 5), including mean ± SD, CV, TE, SWC, and ICC. Resting TSI reported an ICC of 0.91 with a CV of 2.7%, indicating very high within-subject consistency under the standardised laboratory conditions. In contrast, TSI variables derived from the sprint phase demonstrated only moderate reliability (ICC = 0.70) and larger within-subject variability (CV = 20.1%). TSI measures obtained during the recovery phase after Sprint 1 also showed moderate reliability (ICC = 0.64) with a CV of 25.9%, suggesting greater day-to-day fluctuation in these dynamic responses. Time-based variables showed lower ICC values. The duration of the sprint phase had an ICC of 0.44 with a CV of 14.4%, and the duration of the recovery phase after Sprint 1 had an ICC of −0.36 with a CV of 10.4%. These combinations of relatively low CV but poor ICC reflect consistent within-subject measurements alongside notable between-subject differences in these temporal characteristics. Rate of change metrics during the sprint showed an ICC of 0.73 and a CV of 23.9%, whereas recovery rate after Sprint 1 showed an ICC of 0.28 and a CV of 29.1%. Half recovery time yielded an ICC of 0.78 with a CV of 29.3%, indicating that although average values were reasonably consistent at the group level, individual recovery profiles varied between sessions.

Reliability statistics for thigh girth, wellness, and CK obtained from control participants during six repeated assessments are shown in Table 3. Mid-thigh girth demonstrated an ICC of 0.99 and a CV of 0.7%, indicating highly consistent anthropometric measurements. The wellness score showed an ICC of 0.88 with a CV of 7.2%, reflecting acceptable consistency in self-reported wellness assessments. CMJ height also demonstrated high repeatability, with ICC values of 0.97 and a CV of 4.3%. In contrast, creatine kinase (CK) levels displayed moderate reliability (ICC = 0.59) and higher variability (CV = 35.4%), suggesting significant fluctuations in this biochemical marker between sessions. Reliability statistics for repeated cycle sprint performance outcomes are presented in Table 4. Single sprint (Sprint 1 only) outputs, namely Sprint 1 mean power and Sprint 1 peak power, showed high reliability (ICC = 0.96 for both) with low to moderate variability (CV = 6.3% and 5.3%, respectively). Variables derived from all 10 sprints, including total work and best sprint mean power, also demonstrated high repeatability (ICC = 0.97 for both) with low variability (CV = 4.1% to 4.7%). In contrast, fatigue-related indices showed poorer reproducibility, with mean power decrement demonstrating low reliability (ICC = 0.29) and

**Table 2. Summary of reliability statistics of NIRS parameters, including resting, during Sprint 1, immediately following Sprint 1 and Half-recovery time following Sprint 10. Data obtained from control participants (*n* = 5) following six repeated assessments.**

|  | Mean ± SD | CV % [95% CI] | TE [95% CI] | SWC [95% CI] | ICC [95% CI] |
|---|---|---|---|---|---|
| Resting (TSI%) | 73.1 ± 3.4 | 2.7 [1.2; 4.1] | 1.6 [1.2; 2.5] | 0.7 [0.4; 2.0] | 0.91 [0.69; 0.99] |
| **Sprint 1** *Absolute change* |  |  |  |  |  |
| Sprint (ΔTSI%) | −12.4 ± 3.1 | 20.1 [13.4; 26.8] | 2.4 [1.8; 3.8] | 0.6 [0.4; 1.8] | 0.70 [−0.03; 0.97] |
| Recovery (ΔTSI%) | 7.7 ± 3.0 | 25.9 [6.4; 45.5] | 2.2 [1.5; 4.1] | 0.6 [0.4; 1.7] | 0.64 [0.12; 0.95] |
| *Phase duration* |  |  |  |  |  |
| Sprint (s) | 11.3 ± 2.4 | 14.4 [2.7; 26.1] | 2.0 [1.5; 3.1] | 0.5 [0.3; 1.4] | 0.44 [0.05; 0.89] |
| Recovery (s) | 18.1 ± 2.5 | 10.4 [−2.5; 23.3] | 2.8 [2.0; 5.2] | 0.5 [0.3; 1.5] | 0.36 [−0.42; 0.06] |
| *Rate of change* |  |  |  |  |  |
| Sprint (%•s⁻¹) | −1.2 ± 0.4 | 23.9 [9.1; 38.6] | 0.3 [0.2; 0.5] | 0.1 [0.04; 0.2] | 0.73 [0.24; 0.96] |
| Recovery (%•s⁻¹) | 0.4 ± 0.2 | 29.1 [−0.3; 58.5] | 0.2 [0.1; 0.3] | 0.04 [0.02; 0.1] | 0.28 [−0.19; 0.86] |
| **Sprint 10** Half-recovery time (s) | 15.6 ± 5.4 | 29.3 [19.2; 39.3] | 4.8 [3.6; 7.5] | 1.1 [0.7; 3.1] | 0.78 [0.33; 0.97] |

Abbreviations: SD, standard deviation; CV, coefficient of variation; TE, typical error; SWC, smallest worthwhile change; ICC, intraclass correlation coefficient; 95% CI, upper and lower bound 95% confidence interval.

**Table 3. Summary of reliability statistics of thigh girth, countermovement jump, wellness and creatine kinase parameters, obtained from control participants during six repeated assessments.**

| | Mean ± SD | CV [95% CI] | TE [95% CI] | SWC [95% CI] | ICC [95% CI] |
|---|---|---|---|---|---|
| Mid-thigh girth (cm) | 54.4 ± 4.3 | 0.7 [0.4; 1.1] | 0.5 [0.4; 0.9] | 0.9 [0.5; 2.5] | 0.99 [0.99; 1.00] |
| Wellness score | 19.3 ± 2 | 7.2 [5.6; 8.7] | 1.4 [1.1; 2.3] | 0.4 [0.2; 1.2] | 0.88 [0.59; 0.99] |
| CMJ (cm) | 34.8 ± 4.1 | 4.3 [1.0; 7.6] | 1.0 [0.8; 1.6] | 0.8 [0.5; 2.4] | 0.97 [0.91; 0.99] |
| CK (IU) | 187.9 ± 102.5 | 35.4 [8.9; 62.0] | 73.3 [55.2; 115.7] | 20.5 [12.3; 58.9] | 0.59 [-0.33; 0.95] |

Abbreviations: SD, standard deviation; CV, coefficient of variation; TE, typical error; SWC, smallest worthwhile change; ICC, intraclass correlation coefficient; 95% CI, upper and lower bound 95% confidence interval.

high variability (CV = 42.5%), while peak power decrement showed moderate reliability (ICC = 0.70) but similarly high variability (CV = 39.6%), indicating these decrement measures fluctuate substantially between assessments.

Table 5 shows a baseline comparison between the control (CON) and experimental (EXP) groups. The data confirm the absence of significant differences between the two cohorts. Thus, the Typical Error (TE) derived from the CON group provides a robust foundation for interpreting the outcomes in the EXP group post-exercise-induced muscle damage (EIMD).

Turning to the physiological and performance outcomes assessed across the recovery period, despite notable mean changes in some variables, there were no statistically significant differences ($P > 0.05$) over time in mid-thigh girth, perceived recovery, jump height or creatine kinase (CK) levels (Table 6). Similarly, between-group comparisons showed no significant distinctions in how these markers evolved, implying that the intervention did not differentially affect these physiological and performance outcomes. However, when measured changes were compared against the more conservative threshold (TE or SWC, whichever was higher), wellness, CMJ, and CK demonstrated effect sizes ranging from small to extremely large at specific time points (Fig 2). For repeated cycle sprint performance (Table 6), Sprint 1 mean power and Sprint 1 peak power are shown in Fig 2. Across all cycle sprint metrics, changes from baseline displayed 90% confidence intervals that overlapped zero, therefore, standardised effect sizes were considered unclear. Accordingly, there were no significant within-group changes or between-group differences for any cycling sprint outcome ($P > 0.05$).

Time-course comparisons of resting TSI, sprint, recovery, phase duration, rate of change and half-recovery time (Table 7) showed no significant main effects or interactions ($P > 0.05$), implying that, from a conventional null hypothesis perspective, these parameters remained stable across time points and between groups. However, when changes from baseline were evaluated against the pre-specified magnitude-based criteria, several outcomes showed clear deviations.

**Table 4. Summary of reliability statistics of repeated cycle performance obtained from control participants during six repeated assessments.**

| | Mean ± SD | CV [95% CI] | TE [95% CI] | SWC [95% CI] | ICC [95% CI] |
|---|---|---|---|---|---|
| *Single sprint (Sprint 1 only)* | | | | | |
| Sprint 1 mean power (W) | 709.0 ± 104.2 | 6.3 [3.5; 9.0] | 31.5 [23.7; 49.7] | 20.8 [12.5; 59.9] | 0.96 [0.86; 0.99] |
| Sprint 1 peak power (W) | 956.4 ± 117.2 | 5.3 [2.1; 8.6] | 43.0 [32.4; 67.8] | 23.5 [14.1; 67.4] | 0.96 [0.86; 0.99] |
| *Derived from all 10 sprints* | | | | | |
| Total work (kJ) | 40.2 ± 4.5 | 4.1 [2.5; 5.8] | 1.8 [1.4; 2.5] | 1.0 [0.6; 2.8] | 0.97 [0.90; 0.99] |
| Best sprint mean power (W) | 736.4 ± 87.8 | 4.7 [3.3; 6.5] | 29.3 [22.9; 40.7] | 17.0 [10.2; 48.8] | 0.97 [0.91; 0.99] |
| Mean power decrement (%) | 8.5 ± 2.3 | 42.5 [22.6; 67.6] | 4.0 [3.2; 5.6] | 0.4 [0.2; 1.0] | 0.29 [-1.08; 0.90] |
| Peak power decrement (%) | 8.2 ± 3.8 | 39.6 [30.8; 48.7] | 3.4 [2.4; 4.2] | 0.7 [0.4; 2.0] | 0.70 [-0.15; 0.88] |

Abbreviations: SD, standard deviation; CV, coefficient of variation; TE, typical error; SWC, smallest worthwhile change; ICC, intraclass correlation coefficient; 95% CI, upper and lower bound 95% confidence interval.

**Table 5. Comparison of baseline measures between CON and EXP group (mean±SD).**

| | Control (*n*=5) | EXP (*n*=12) | *P* value |
|---|---|---|---|
| Body mass (kg) | 77.6±7.2 | 76.6±10.8 | 0.85 |
| PPO (W) | 313.9±22.5 | 312.8±42.6 | 0.96 |
| $VO_{2max}$ (ml$^{-1}$ kg$^{-1}$ min$^{-1}$) | 47.8±4.1 | 46.9±6.7 | 0.79 |
| 1RM (kg) | – | 112.6±30.8 | – |
| Mid-thigh skinfold (cm) | 12.2±7.3 | 13.2±2.3 | 0.76 |
| Resting TSI (%) | 71.7±4.5 | 67.7±6.0 | 0.20 |
| Mid-thigh girth (cm) | 54.3±4.2 | 54.2±2.6 | 0.98 |
| Wellness score | 19.7±1.4 | 18.1±1.9 | 0.11 |
| CMJ (cm) | 34.2±3.6 | 34.9±6.6 | 0.84 |
| CK (IU) | 225.3±132.6 | 146.3±25.3 | 0.25 |
| Sprint 1 mean power (W) | 720.1±83.2 | 666.8±100.0 | 0.31 |
| Sprint 1 peak power (W) | 982.6±120.3 | 953.1±144.9 | 0.70 |
| Total work (kJ) | 41.0±4.9 | 38.3±5.3 | 0.35 |
| Best sprint mean power (W) | 746.0±85.0 | 691.1±94.0 | 0.28 |
| Mean power decrement (%) | 8.5±1.8 | 7.6±4.5 | 0.66 |
| Peak power decrement (%) | 9.6±3.4 | 9.0±7.8 | 0.86 |

Abbreviations: PPO, peak power output during incremental cycle test; $VO_{2max}$, maximal aerobic capacity; RM, repetition maximum; CMJ, countermovement jump; CK, creatine kinase.

**Table 6. Baseline and change from baseline (Δ) wellness, CK, countermovement and repeated cycle sprint performance of the EXP group following EIMD.**

| | Baseline | Δ Post 1 h | Δ Post 24 h | Δ Post 48 h | Δ Post 72 h | Δ Post 96 h |
|---|---|---|---|---|---|---|
| Mid-thigh girth (cm) | 54.2±4.2 | −0.1±0.5 | 0.2±0.5 | 0.1±0.7 | 0.3±0.6 | −0.1±0.6 |
| Wellness score | 18.1±1.9 | −2.3 [L] ±2.4 | −1.25±2.7 | −1.8[VL]±3.0 | −1.3±3.4 | 0.1±2.7 |
| CMJ (cm) | 34.9±6.6 | −3.5[S] ± 3.9 | −3.71[M]±4.9 | −4.0[M]±6.3 | −3.1[S] ± 5.5 | −1.4±3.9 |
| CK (IU) | 146.3±25.3 | 150.9[EL] ± 107.1 | 1884.0 [EL] ± 1430.0 | 1364.6 [L] ±1797.6 | 846.1[M] ±1296.4 | 583.6[M]±963.7 |
| *Single sprint (Sprint 1 only)* | | | | | | |
| Sprint 1 mean power (W) | 666.8±100.0 | −35.9±109.9 | −49.56±114.5 | −55.3±108.4 | −4.78±65.0 | 20.5±65.9 |
| Sprint 1 peak power (W) | 953.1±144.9 | −52.2±154.4 | −55.8±151.4 | −58.0±158.1 | 4.7±92.8 | 23.0±86.1 |
| *Derived from all 10 sprints* | | | | | | |
| Total work (kJ) | 38.3±5.3 | −2.0±4.5 | −2.3±5.3 | −2.4±5.8 | −0.4±3.3 | 0.3±3.0 |
| Best sprint mean power (W) | 691.1±94.0 | −32.8±82.9 | −33.3±87.4 | −49.6±98.6 | −9.5±51.8 | 10.6±53.9 |
| Mean power decrement (%) | 7.6±4.5 | 0.6±3.4 | 1.3±4.0 | −0.8±3.1 | −0.1±2.5 | 0.8±3.2 |
| Peak power decrement (%) | 9.0±7.8 | −0.5±5.4 | −0.5±5.7 | −1.3±5.5 | −2.4±5.5 | −1.0±6.7 |

Abbreviations: CMJ, countermovement jump; CK, creatine kinase. Δ = difference between timepoint minus baseline. [S, M, L, VL, EL] indicate small, moderate, large, very large or extremely large effect sizes, respectively.

Specifically, resting TSI at 24 h post-EIMD was interpreted as a small but clear increase because the mean change exceeded the TE derived from control participants (TE 1.6 TSI%, which is larger than the SWC of 0.7 TSI%) and the 90% CI for the change excluded zero (Fig 3). In addition, both recovery after Sprint 1 (ΔTSI%) and rate of change (%•s$^{-1}$) at 1 h post-EIMD demonstrated large standardised effects (Figs 4 and 5) consistent with changes that exceeded the TE thresholds. Collectively, these observations highlight potentially meaningful alterations that are not captured by *p*-values

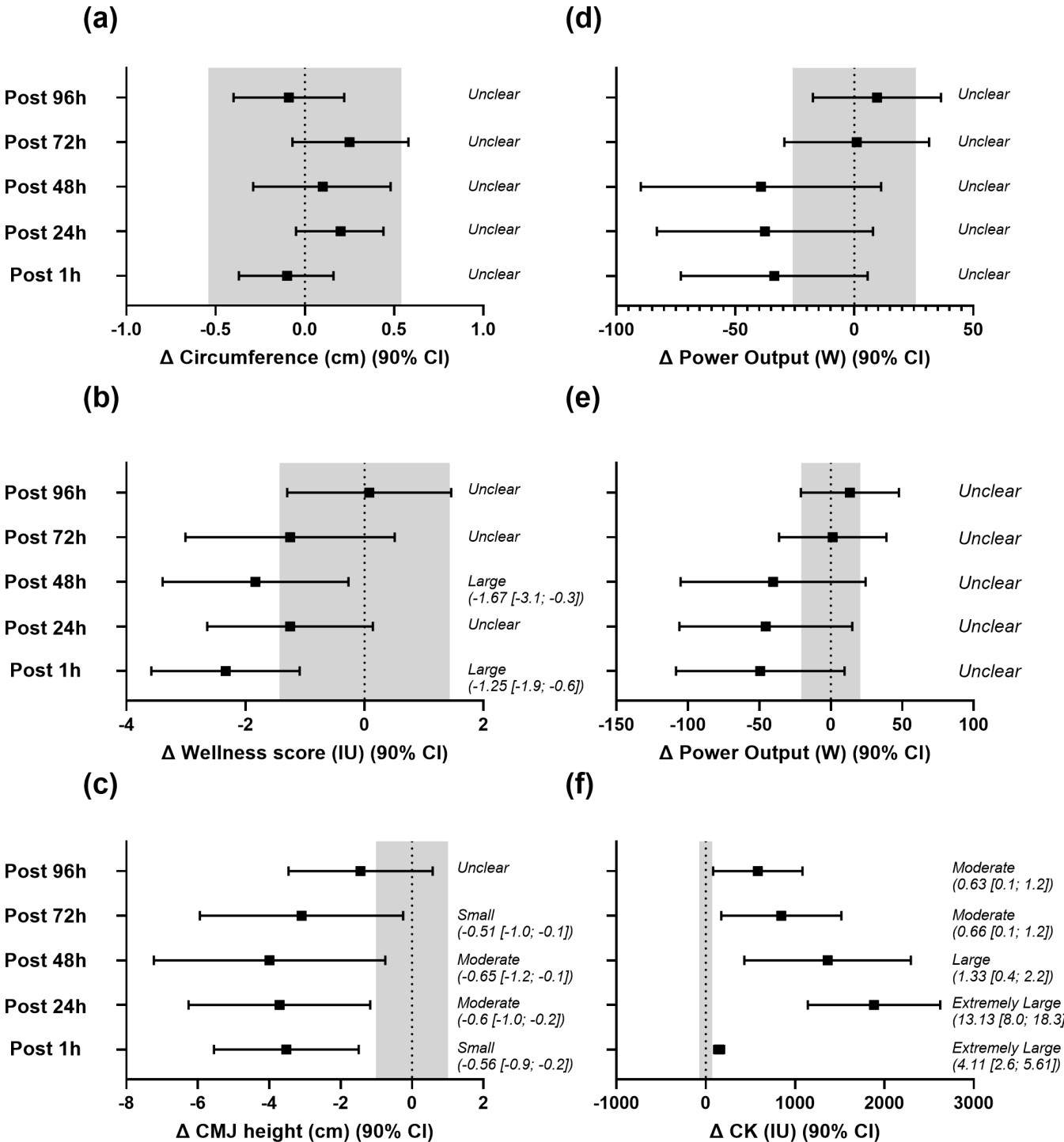

**Fig 2. Changes from baseline (Δ) in (a) mid-thigh girth, (b) perceived wellness, (c) countermovement jump height, (d) cycle sprint 1 mean power, (e) cycle sprint 1 peak power, and (f) creatine kinase concentration in the experimental (EXP) group.** Values are presented as means with 90% confidence intervals (CI). The shaded region represents the greater of either the typical error (TE) or the smallest worthwhile change (SWC), used to interpret practical significance. Outcomes are classified as unclear when the 90% CI overlaps both zero and the shaded region. Standardised effect sizes (±90% CI) are reported for outcomes exceeding this threshold.

**Table 7. Baseline and change from baseline (Δ) NIRS variables of the EXP group following EIMD (mean±SD).**

| | Baseline | Δ Post 1 h | Δ Post 24 h | Δ Post 48 h | Δ Post 72 h | Δ Post 96 h |
|---|---|---|---|---|---|---|
| Resting TSI (%) | 67.7±6.0 | 0.7±6.3 | 2.1[S] ± 3.9 | 1.6±5.4 | 1.4±3.5 | 1.3±3.2 |
| **Sprint 1** | | | | | | |
| *Absolute change* | | | | | | |
| Sprint (TSI%) | −17.5±5.2 | 4.0[L]±6.3 | −2.0±5.8 | −0.5±5.9 | −0.4±3.8 | −1.2±4.1 |
| Recovery (TSI%)* | 8.6±4.2 | 0.5±2.8 | −0.4±3.1 | −0.2±6.0 | 0.8±3.3 | 0.5±3.1 |
| *Phase duration* | | | | | | |
| Sprint (s) | 11.0±1.5 | 0.3±1.8 | 0.7±1.7 | −0.1±1.7 | 0.3±1.6 | 0.2±1.9 |
| Recovery (s)* | 19.3±3.4 | −1.0±2.9 | −0.4±1.1 | −2.1±7.1 | −4.5±9.2 | −2.8±6.7 |
| *Rate of Change* | | | | | | |
| Sprint (%•s⁻¹) | −1.62±0.5 | 0.38 [L]±0.6 | −0.12±0.6 | −0.02±0.5 | 0.03±0.3 | −0.06±0.4 |
| Recovery (%•s⁻¹)* | 0.45±0.2 | 0.05±0.1 | −0.01±0.2 | −0.29±0.9 | −0.35±1.0 | −0.30±1.1 |
| **Sprint 10** Half-recovery time (s)* | 19.9±8.4 | −3.0±7.8 | −2.64±8.1 | −2.1±12.4 | 0.9±21.9 | −2.9±11.5 |

Δ=difference between time points minus baseline. *indicates n=11 due to erroneous data from one participant. [S, M, L, EL] indicate small, moderate, large or extremely large effect sizes, respectively.

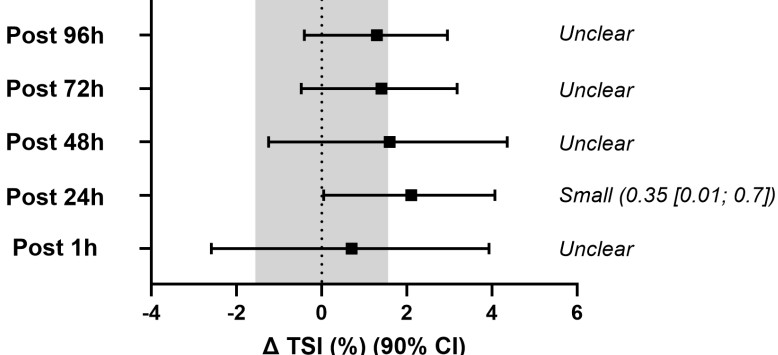

**Fig 3. Changes in resting tissue saturation index (ΔTSI%) from baseline in the experimental (EXP) group.** Values are presented as means with 90% confidence intervals (CI). The shaded region represents the greater of either the typical error (TE) or the smallest worthwhile change (SWC), serving as a threshold for interpreting practical significance. Outcomes are classified as unclear when the 90% CI overlaps zero and the shaded region. Standardised effect sizes (±90% CI) are reported for time points where changes exceed this threshold.

alone. From a practical standpoint, the presence of large effect sizes, despite nonsignificant ANOVA results, emphasises the value of supplementing null hypothesis tests with magnitude-based interpretations, particularly when dealing with smaller samples or high inter-individual variability.

## Discussion

This study primarily aimed to determine whether NIRS–derived indices, measured at rest and during dynamic exercise, are sensitive to potential alterations in muscle oxygenation associated with exercise-induced muscle damage and recovery status. Although conventional statistical analyses did not reveal significant differences over time or between groups, the magnitude of the observed effect sizes, particularly those associated with elevated creatine kinase, reduced self-reported wellness, and reduced countermovement jump performance, highlights the practical relevance of these variables as markers of muscle damage and its functional consequences. These markers were included to verify that the eccentric squat protocol induced a meaningful EIMD

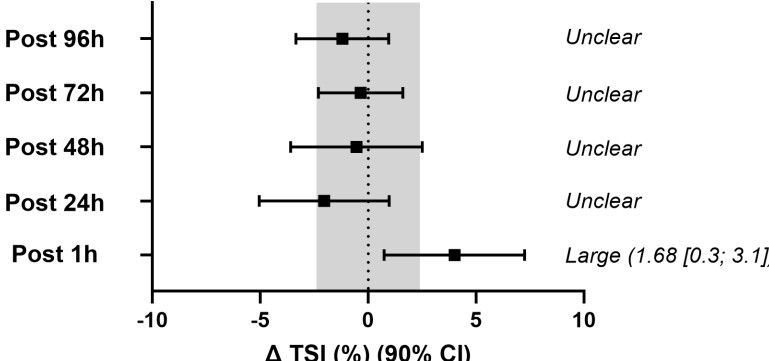

**Fig 4. Change in tissue saturation index (ΔTSI%) during Sprint 1 from baseline in the experimental (EXP) group.** Values are presented as means with 90% confidence intervals (CI). The shaded region represents the greater of either the typical error (TE) or the smallest worthwhile change (SWC), serving as a threshold for interpreting practical significance. Outcomes are classified as unclear when the 90% CI overlaps zero and the shaded region. Standardised effect sizes (±90% CI) are reported for time points where changes exceed this threshold.

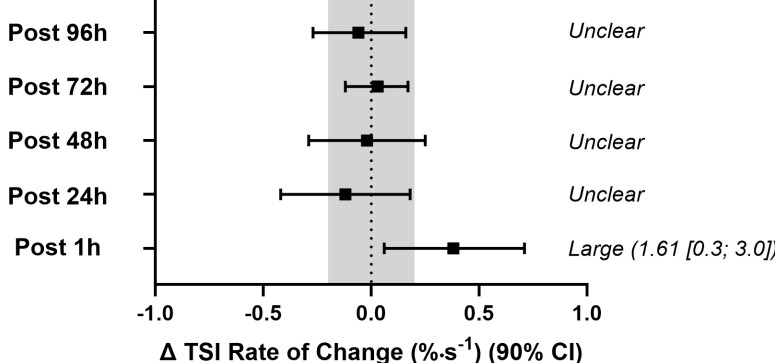

**Fig 5. Change in the rate of tissue saturation (%·s⁻¹) during Sprint 1 from baseline in the experimental (EXP) group.** Values are presented as means with 90% confidence intervals (CI). The shaded region represents the greater of either the typical error (TE) or the smallest worthwhile change (SWC), which serves as the threshold for interpreting practical significance. Outcomes are classified as *unclear* when the 90% CI overlaps both zero and the TE region. Standardised effect sizes (±90% CI) are reported where changes exceed this threshold.

response, and the NIRS outcomes were then used to characterise any accompanying changes in local oxygenation. We therefore interpret the NIRS findings as describing how oxygen delivery and utilisation behave in a muscle that is known to be damaged, rather than as evidence that impaired oxygen availability directly explains the observed decline in jump performance.

Several studies have utilised markers such as CK, self-reported wellness, and performance measures (e.g., CMJ) as indirect indices of eccentric EIMD [1–3,5]. In the present study, increases in CK values and reductions in wellness scores and CMJ performance did not reach statistical significance; however, the associated effect sizes ranged from small to extremely large across multiple time points. This supports the interpretation that the EIMD protocol induced physiologically meaningful muscle perturbations.

Cycle sprint performance was included to characterise the exercise performance capacity of the cohort and to provide a practical functional context for interpreting the NIRS responses. In the present study, neither Sprint 1 outputs nor the set level cycle metrics demonstrated clear changes across the recovery period, with changes from baseline characterised by confidence intervals overlapping zero and no evidence of between-group differences. Reliability analyses indicated that

absolute output metrics, including Sprint 1 mean and peak power, total work, and best sprint mean power, were highly reproducible, whereas decrement indices were notably more variable between repeated assessments. Collectively, these findings suggest that the eccentric squat protocol elicited perceptual, biochemical and jump-based consequences of muscle damage, yet did not translate into a detectable impairment in repeated sprint mechanical output within this brief cycling task, and that fatigue indices derived from repeated sprints may be less stable markers of change than absolute power or work in this context.

Our NIRS data demonstrate that resting TSI is highly reproducible, with low CV and excellent ICCs, and that it showed a small but consistent upward shift (~2%) at 24 hours post EIMD compared with baseline. While this pattern was not supported by conventional null hypothesis testing, the mean change exceeded the conservative TE threshold derived from control participants (which was larger than the SWC) and the 90% confidence interval for the change excluded zero, supporting interpretation as a small but clear change rather than measurement noise. In contrast, dynamic NIRS variables during the sprint phase were more variable, yet effect size analyses revealed large changes in specific measures at 1 hour post EIMD, particularly the magnitude and rate of TSI desaturation during a 6-second sprint (Table 6). Taken together, this pattern is consistent with the view that eccentric exercise-induced muscle damage can disturb the balance between oxygen delivery and utilisation in skeletal muscle during contractions. Studies using NIRS and ultrasound have shown that EIMD can slow the onset of muscle deoxygenation, increase the ratio of oxygen delivery to utilisation during severe intensity exercise [11], and impair indices of microvascular reactivity [14], suggesting that damaged muscle extracts less oxygen for a given convective $O_2$ supply. In this context, the blunted desaturation and slower desaturation rate we observed at 1 hour may reflect a relative increase in $O_2$ availability for a given metabolic demand, driven by a combination of altered motor unit recruitment, muscle oedema and local vascular dysfunction in the damaged tissue [19,45]. Conversely, the small upward shift in resting TSI at 24 hours aligns with previous reports of elevated resting muscle TSI for 24–48 hours or longer after eccentric exercise or downhill walking [12], which have been attributed to reduced muscle oxygen consumption, increased local blood flow, or both [46]. Such changes are thought to reflect a combination of reduced oxidative demand in damaged fibres and residual local hyperaemia driven by vasodilatory metabolites and mediators (e.g., nitric oxide, prostaglandins and histamine) and by the inflammatory response to muscle damage [14,47]. Within this EIMD model, the absence of marked changes in TSI half recovery time suggests that the capacity to re-establish saturation during passive recovery is largely preserved, and that EIMD primarily influences the balance between oxygen delivery and utilisation during contractions rather than the speed of reoxygenation once exercise ceases [12,14].

Our findings reveal both agreement and divergence with earlier studies, which may be attributable to differences in EIMD protocols, NIRS measurement methodologies, and the timing of assessments. Prior research demonstrating changes in NIRS variables following EIMD generally employed protocols involving substantial eccentric loading coupled with either prolonged exercise or occlusion-based measurements. For example, slowed [HHb] kinetics during cycling have been observed following eccentric squat protocols, suggesting impaired matching of oxygen delivery to utilisation, possibly indicative of microvascular dysfunction secondary to muscle damage [11]. Comparable studies using downhill walking combined with comprehensive measurement protocols including rest, arterial and venous occlusions, and submaximal isometric contractions at 30%, 50%, and 80% of maximal voluntary contraction, have reported transient increases in resting tissue oxygen saturation approximately 30 minutes post-EIMD, which typically return to baseline within 24 hours [12]. In contrast, our study adopted a more practical, non-invasive approach by assessing resting TSI and capturing dynamic desaturation during a brief 6-second cycle sprint, without the need for vascular occlusions. These methodological and temporal differences highlight how exercise modality and measurement technique can influence the detection of post-EIMD oxygenation changes.

Conversely, other studies have reported no significant alterations in NIRS-derived indices following muscle-damaging protocols. In one such investigation, EIMD was induced via 30 minutes of high-intensity eccentric cycling. Despite the

 

use of arterial occlusion to assess oxyhaemoglobin kinetics both during and after the intervention, no significant changes were detected, suggesting that mitochondrial oxidative capacity may remain largely unaffected under these specific conditions [13]. Similarly, electrically stimulated knee extensions used to provoke muscle damage have been shown to elicit no meaningful changes in TSI%, tHb, $O_2$Hb, or HHb, despite reductions in critical torque [20]. Notably, assessments in this latter study were limited to pre- and 48-hour post-intervention time points, and data were collected during a 5-minute critical torque test, contrasting with our approach, which evaluated TSI at rest and during a 6-second sprint following 100 eccentric squats. Differences in exercise stimulus, duration of eccentric loading, and NIRS measurement protocols (occlusion-based versus rapid, field-compatible assessments), almost certainly contribute to the disparate NIRS responses reported in the EIMD literature and should be considered when comparing across studies. While studies imposing prolonged eccentric loading or incorporating occlusion techniques during NIRS measures [11,12] tend to reveal pronounced changes in muscle oxygenation, our more practical and rapid approach yielded dynamic alterations in sprint desaturation measures at 1-hour post-EIMD. Taken together, these comparisons indicate that EIMD-related alterations in muscle oxygenation can be detected with NIRS, but the magnitude and timing of detectable changes depend on the damaging stimulus and the exercise task used to interrogate oxygen delivery and utilisation. In the present study, a brief, field-compatible approach identified small to large changes in selected indices at specific time points after EIMD, whereas several other NIRS variables showed no clear change. This suggests that short practical assessments may capture some perturbations in local oxygenation after eccentric exercise, but may be less sensitive than protocols that use a more metabolically demanding exercise task, or that incorporate occlusion-based measures.

An important conceptual consideration emerging from these comparisons is whether the eccentric exercise-induced muscle damage (EIMD) protocol used in this study (unaccustomed eccentric squats combined with short-duration cycle sprints) elicits sufficient disruption in muscle oxygenation to be detected by NIRS. Although our protocol elicited clear biochemical (elevated CK), perceptual (reduced wellness), and performance (impaired CMJ) markers of muscle damage, its predominantly mechanical and strength-focused nature may not perturb oxygen delivery and utilisation as significantly as exercise modalities with a substantial aerobic component. For example, prolonged downhill running, which imposes continuous eccentric loading under weight-bearing conditions, can exacerbate microvascular dysfunction and oxidative stress, thereby increasing metabolic demand and potentially leading to more sustained alterations in muscle oxygenation. Similarly, high-intensity intermittent exercise (HIIT) imposes significant oxidative and metabolic stress, which may amplify the effects of muscle damage on oxygen dynamics. These insights suggest that alternative exercise protocols that incorporate a higher aerobic or metabolic strain may offer enhanced sensitivity for NIRS measurements, thereby providing deeper insights into the interplay between structural muscle damage, microvascular dysfunction, and oxygen utilisation.

Methodological factors, including scale dependency and data processing strategies, can also affect the sensitivity and reliability of NIRS-derived measures. Our results show that measurements obtained on a larger absolute scale, such as resting TSI, exhibit inherently lower relative variability compared to dynamic delta measures. Resting TSI, recorded at relatively high mean values, displayed a low CV (2.7%) and excellent reproducibility (ICC = 0.91). In contrast, dynamic delta measures, reflecting smaller absolute changes in TSI during exercise, demonstrated higher CV values despite similar absolute measurement errors. Consequently, the high reproducibility of resting TSI is partly a function of this scale dependency. As such, direct comparisons between static and dynamic NIRS measures should be made cautiously. Our study also highlights practical considerations in data processing, particularly the use of filtering techniques to enhance the interpretability of NIRS signals. The application of a zero-phase shift, low-pass Butterworth filter was used to attenuate high-frequency noise and produce a smoother signal trajectory during the sprint and recovery epochs. In the present analysis, we did not directly compare filtered with unfiltered data; therefore, we cannot determine whether filtering increased the sensitivity of NIRS outcomes to EIMD. However, the filtered traces provided stable and physiologically plausible desaturation profiles from which effect sizes could be derived, and we consider this approach a reasonable compromise between noise reduction and preservation of the underlying signal when NIRS is used in brief, high-intensity protocols.

## Limitations

Several limitations of this study warrant consideration. The control group comprised only five participants, which was used to establish typical error (TE) values for comparison with the EIMD intervention. A larger control sample may have reduced the TE and provided more precise estimates of variability, thereby potentially revealing additional variables of practical significance [41]. Second, CMJ depth was self-selected rather than fixed by joint angle or centre-of-mass displacement. Although this approach is consistent with many CMJ monitoring and reliability protocols [48], variation in self-selected depth may have interacted with soreness-related changes in joint stiffness and jump strategy, such that participants subtly adjusted countermovement mechanics as exercise-induced muscle damage and delayed-onset muscle soreness developed, potentially masking small changes in neuromuscular function [49]. Future studies could consider angle or displacement standardised CMJ procedures where tighter control of jump mechanics is required. The repeated sprint protocol employed in this study may have confounded the recovery process. The daily repeated sprints could have introduced cumulative fatigue and additional muscle damage, thereby affecting both recovery markers and NIRS-derived parameters, for example, by altering baseline fatigue state, motor unit recruitment patterns or local perfusion. This repeated exposure might have blurred the distinction between the EIMD and control groups over the testing period, as the recovery process was potentially compromised by cumulative exercise stress [50]. Evidence from comparable protocols involving short-duration maximal efforts (8 x 5-second efforts) with active recovery has shown no significant functional impairments 24 hours post-exercise [51]. Given the similarity to our 10 × 6-second repeated sprint protocol, this is unlikely to have induced additional muscle stress that influenced the NIRS outcomes, particularly as both groups completed the same sprint work. Future work with larger samples and protocols specifically designed to interrogate repeated-sprint dynamics could extend the present approach by modelling NIRS responses across all sprints; however, this was beyond the scope and primary aims of the current study. In addition, future studies that monitor recovery over several days may wish to minimise the volume and frequency of maximal-effort tests, or to separate them temporally from key NIRS assessments, to reduce the risk that the assessment protocol itself delays or alters the underlying recovery trajectory.

Our protocol did not include the collection of NIRS variables during prolonged submaximal exercise. This decision was made to maintain a practical, rapid assessment approach; however, it may have limited the detection of more nuanced changes in muscle oxygenation during sustained activity. Moreover, we did not implement arterial occlusion techniques to assess muscle oxygen consumption ($mVO_2$) and muscle blood flow (mBF), which have been shown in other studies [12] to provide further insight into microvascular function and recovery. Future studies employing such techniques may help to elucidate the full potential of NIRS as a tool for monitoring recovery following exercise-induced muscle damage.

Differences in muscle fibre type composition between muscle groups may have influenced the NIRS-derived oxygenation kinetics observed in this study. The gastrocnemius has a relatively greater proportion of slow-twitch oxidative fibres compared to the vastus lateralis [52], which may alter the pattern of local oxygen utilisation and reoxygenation following eccentric exercise. Given the preferential disruption of fast-contracting fibres during eccentric loading [53], there may be a compensatory shift in motor unit recruitment toward slower, more oxidative fibres during subsequent efforts. This shift is likely to result in elevated muscle oxygen consumption for a given task, potentially leading to faster or more pronounced desaturation and a different reoxygenation profile during recovery [46]. As NIRS captures the balance between oxygen delivery and utilisation, the underlying fibre composition and post-damage recruitment strategy are important contextual factors that can influence both the magnitude and temporal characteristics of the oxygenation signal. These considerations highlight the importance of muscle site selection when using NIRS to assess recovery from eccentric-induced muscle damage. Finally, although mid-thigh skinfold was low in this sample and within recommended limits relative to the 35 mm inter-optode distance, we did not explicitly adjust NIRS variables for adipose tissue thickness; given that ATT can influence the amplitude of NIRS signals, the present findings are most directly applicable to similarly lean, physically active populations.

These limitations suggest that while our findings provide valuable insights into the sensitivity of NIRS-derived measures following EIMD, future research with larger control samples, refined testing protocols that minimise cumulative fatigue, and enhanced methodological control is warranted.

## Conclusion

In this cohort of healthy, physically active adults, resting tissue saturation index (TSI) measured with NIRS showed excellent reproducibility across repeated days, and selected NIRS-derived indices exhibited small but systematic changes in temporal association with an eccentric squat protocol that elicited biochemical, perceptual and functional markers of exercise-induced muscle damage. Although traditional statistical tests did not identify significant group-by-time effects, effect size analyses indicated a modest upward shift in resting TSI at 24 hours post EIMD and large changes in the magnitude and rate of TSI desaturation during a 6-second sprint at 1 hour. These observations suggest that NIRS measures may have the potential to contribute to non-invasive characterisation of muscle status during recovery from unaccustomed eccentric exercise; however, the present findings should be regarded as preliminary. Larger, hypothesis-driven studies are required to better characterise how NIRS-derived indices relate to the severity and time course of muscle damage and to establish whether, and in what form, these measures can be integrated into applied athletic or clinical monitoring.

## Supporting information

**S1 File. PixelRecog.m.** Custom MATLAB script used to extract mean±SD values from image-based figures reporting muscle oxygen saturation data. The script allows calibration against a known Y-axis scale (e.g., TSI%), manual selection of mean data points and error bars, and outputs oxygen saturation values and standard deviations. This tool facilitates the digitisation of published figures where raw data are unavailable, enabling use in meta-analyses or retrospective power calculations. (TXT)

## Acknowledgments

We would like to express our sincere gratitude to all the participants of this study.

## Author contributions

**Conceptualization:** Chris McManus, Chris Cooper.

**Data curation:** Chris McManus.

**Formal analysis:** Chris McManus.

**Funding acquisition:** Chris Cooper.

**Investigation:** Chris McManus, Kelly Murray, Elizabeth Welbourn, Julie Double.

**Methodology:** Chris McManus, Kelly Murray, Chris Cooper.

**Project administration:** Chris McManus, Kelly Murray, Elizabeth Welbourn, Julie Double.

**Resources:** Chris McManus, Kelly Murray, Elizabeth Welbourn, Julie Double, Chris Cooper.

**Software:** Chris McManus.

**Supervision:** Chris McManus, Chris Cooper.

**Validation:** Chris McManus, Kelly Murray, Elizabeth Welbourn, Julie Double, Henry Chung, Sally Waterworth, Ben Jones, Chris Cooper.

**Visualization:** Chris McManus.

**Writing – original draft:** Chris McManus.

**Writing – review & editing:** Chris McManus, Kelly Murray, Elizabeth Welbourn, Julie Double, Henry Chung, Sally Waterworth, Ben Jones, Chris Cooper.

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
