## [Decision Letter · Decision Letter 0]

11 Sep 2025

Dear Dr. McManus,

We look forward to receiving your revised manuscript.

Kind regards,

Laurent Mourot

Section Editor

PLOS ONE

Journal Requirements:

3. We are unable to open your Supporting Information file “PixelRecog.m”. Please kindly revise as necessary and re-upload.

Reviewers' comments:

Reviewer's Responses to Questions

**Comments to the Author**

1. Is the manuscript technically sound, and do the data support the conclusions?

Reviewer #1: Partly

Reviewer #2: Yes

2. Has the statistical analysis been performed appropriately and rigorously?

Reviewer #1: Yes

Reviewer #2: Yes

3. Have the authors made all data underlying the findings in their manuscript fully available?

Reviewer #1: Yes

Reviewer #2: Yes

4. Is the manuscript presented in an intelligible fashion and written in standard English?

Reviewer #1: Yes

Reviewer #2: Yes

Reviewer #1: This study aims to characterize the effect of exercise-induced muscle damage (EIMD) on muscle function and muscle oxygenation indices, evaluated through near infrared spectroscopy (NIRS). The goal is to establish the validity of NIRS as a practical tool for monitoring EIMD in field settings and inform athletes for recovery strategies. Seventeen participants were recruited and divided into a control group (n=5) and an experimental group (n=12). The control group was used to establish the reliability of the measurement at rest using 6 repetitions of baseline evaluations (Day 0, Day 0+1h, Day 1, Day 2, Day 3 and Day 4) and did not perform muscle damaging exercises. The experimental group did the first baseline evaluation and then perform a muscle damaging exercise based on 10x10 barbell squats at 80% of concentric 1RM. Baseline evaluations were also repeated at (Day 0+1h, Day 1, Day 2, Day 3 and Day ). Results from the control group provided evaluation and reliability estimates of the parameters (typical error and smallest worthwhile change thresholds). No statistically significant time effect nor between group differences were found. However, effect size-based inferences demonstrated small to extremely large effect size in CMJ, wellness, CK and key NIRS parameters. The extent to which EIMD were really induced in this protocol is not clear in the manuscript as no statistically significant reduction in muscle function parameters are reported (countermovement jump, peak and mean sprint power). How the results reported here actually inform the athletes for recovery strategies is not clear. The authors might consider the comments below to help the reader better understanding the novelty and practical implication of the results.

Major comments

1/ L108: Aims of the study: it is not clear why max testing is more convenient than cuff occlusion.

2/ L127: why was a moderate reliability (ICC=0.5) used to estimate sample size? We understand that this allows lowering the required number of participants but it also reduces the strength of the data collected.

3/ L135: the same remark also applies to the 80% power used to estimate the sample size of the experimental group.

4/ the length of the vertical push is key in jumping performance. Therefore the self-chosen depth used for CMJ assessment is surprising, especially in this kind of setting including repeated measurements over time. How can this methodological issue have impacted on the results? (limitations?)

5/ L220: how was the linear factor for sprint tests individually determined exactly ?

6/ the recovery kinetics of tissue saturation index were modelled using a single exponential function. It is not clear how these kinetic parameters were used (not presented) and how the half-recovery time was calculated.

7/ How was mid-thigh skinfold used in NIRS data analysis and interpretation-?

8/ Data used to compute the wellness score should be presented, especially the DOMS score, very relevant in this particular study.

8/ The extent of EIMD is not easy to evaluate in this study. No significant muscle swelling (mid-thigh girth), no DOMS score provided, no change in peak and mean sprint power and no significant change in CMJ. Effect size for CK difference was moderately to extremely large but with very large interindividual variability. This suggest that some participants might have experienced EIMD while some others did not. Would it be possible to identify subgroups (EIMD vs no EIMD) in your population?

9/ Little data are presented from the repeated sprint test (only peak and mean power). Could the authors also include the reduction in peak or mean power from the 1st to the 10th ? What about total mechanical work over the whole repeated sprint test?

10/ Did the authors try to correlate functional parameters (CMJ and sprint tests) with NIRS indices? This could help establish how NIRS-related indices could help monitoring EIMD.

11/ Based on the results presented in table 6, the main message is: 1/ the absolute level of muscle deoxygenation and the rate of muscle deoxygenation tends to be reduced 1h after 10x10 barbell squat at 80% 1RM (large effect size), 2) resting muscle oxygenation tends to be is increased 24h post 10x10 barbell squat at 80% 1RM. No oxygenation changes were observed during recovery after the first (absolute or rate of change) or the 10th sprint (half recovery time). Could the authors discuss better the potential mechanisms underlying these observations and how they compare to previous literature in the field ?

12/ End of 1st paragraph in discussion: “..suggesting that compromised muscle function following eccentric exercise is at least partly underpinned by perturbations in oxygen delivery or utilization”. This sentence is difficult to follow based on your data, because the only functional test that tends to be impaired after eccentric exercise is CMJ, which is not known to be O2 dependent….

13/ End of 3rd paragraph in discussion: “… whereby local blood flow to the exercising limb remains elevated during the early recovery period…” What could the mechanisms behind this ?

14/ Only the first sprint of the repeated sprint test was analysed for deoxygenation and reoxygenation magnitude and kinetics. What about the other sprints? Is only the first sprint informative ? Could the NIRS responses be even more significant later on during the test?

15/ End of 5th paragraph: “…yet pronounced alteration in resting muscle oxygenation…” This statement is very strong as resting TSI did not demonstrate any statistically significant changes and only increased by about 3%.

16/ “…suggest that filtering may enhance the capacity of NIRS to detect physiologically meaningful changes in muscle oxygenation following IEMD.” This statement would require a comparison of NIRS data analysed with vs without filtering.

17/ Limitations: how repeated sprint actually impacted on muscle oxygenation responses?

18/ Conclusion is not fully supported by the present data. For instance: “this study demonstrates that NIRS-derived indices are sensitive to muscle oxygenation perturbations following exercise-induced muscle damage.” NIRS was used to monitor muscle oxygen status, so you observed changes in NIRS-derived indices but to what extent it actually relates to modifications in muscle oxygenation or muscle damage remains unclear. Correlation analyses could help in this regard.

19/ Conclusion: “…for monitoring recovery and underscore its potential application in both athletic and clinical settings”. NIRS-derived parameters were not statistically significantly altered post eccentric exercises and only effect-size inferences indicate potential meaningful changes at 1h et 24h post exercise. It is not clear how to interpret these findings in terms of recovery from muscle damage. Does that mean possible muscle damage fully recovered at 48h post eccentric exercise? Potential applications in both athletic and clinical settings are mentioned several times throughout the manuscript but it is not clear how the authors actually see them?

Minor comments

1/ L75: “Similarly…” this sentence (acceleration of oxygen desaturation) seems to indicate the opposite than the previous one (slow deoxygenation). Need to adapt ?

2/ L89: these effects are not necessarily observed. This sentence should indicate unaccustomed or high-intensity eccentric exercises can lead to muscle damage

3/ L99: “invasive vascular occlusion…” to what methodology do the authors refer to ? Cuff inflation is not invasive

4/ L153: why was stretching “optional” ?

5/ L157: a ref is lacking for VO2max criteria

6/ L164: a 5s time window seems rather short to determine accurate VO2max data. Usually 30s windows are preferred. Could the authors include a ref supporting the 5s time window they used?

7/ L190: anthropometry

8/ L315: “…calculating 0.2 of the between-participant…” Is there a typo error here ?

9/ L356: “fair variability…” for consistency, wouldn’t it be better to use “moderate variability” ?

10/ Figure 4 and figure 5: please indicate the data were collected at sprint 1

11/ “highlight practical alterations in oxygen delivery and utilization associated with muscle damage”. It may be more accurate to mention “alterations in O2 delivery/utilization balance”.

Reviewer #2: Introduction

The introduction is well written and provide a clear rational to understand the need to develop an on-field test to track EIMD. However, the rational about the testing procedure (i.e. sprinting exercise) remain undiscussed, which could limit its implementation. I suggest the authors to strengthen the use of sprinting exercise while we can consider that repetitive CMJ can provide similar findings about oxyhemoglobin desaturation / resaturation kinetics, with a greater precision about muscular work provided, and therefore to infer on a work-to-substrate ratio.

Methods:

L 114: could the authors precise what they define by recreationally active male (for instance; training background, weekly training volume, type of activity)? Were all the participants naïve from eccentric solicitations during their physical activities?

L 120: could the authors report the identification code provided by the UE Research Ethics Committee?

L 126-128: could the authors precise which variable they consider precisely to determine the sample size of the control group (as done for the experimental group)?

L 139: what was done during the familiarization session? Were the participants accustomed to the testing procedure and/ or exercise protocol? Whenever they would familiarize to exercise protocol, how did the experimenter account for induction of neuromuscular fatigue and its recovery, or alternatively, for EIMD?

L 221: what was the level / intensity of this standardized resistance? Were the participants allowed to stand-up from the saddle when sprinting?

L 223-224: could the authors precise which power output values were provided by the bicycle ergometer (i.e. maximal, average power over a revolution, other?)? What was the sampling frequency from the Lode?

Discussion

In my opinion, the first paragraph of the discussion section lacks of relevancy regarding the findings of your study. Specifically, markers of EIMD that you mention are still accepted in the literature to monitor the level of muscular disturbance and performance alteration, so your study only confirm, at most, previous knowledge. That said, I do not understand why these markers are presented at the beginning of this section, while there was no mention of those variables in the objectives or hypothesis.

Furthermore, I do not question whether oxygen delivery is impaired following eccentric exercises, but in your study, I am not convinced that reduction in CMJ performance could be explain by reduction in oxygen delivery. Rather, I guess that alteration in CMJ performance would be explain by EIMD and alteration in excitation-contraction coupling process (e.g. calcium release, sensitivity of stretching reflex or perturbations of sarcomeres’ integrity) that would hinder maximal force production capacity.

I suggest the authors to reconsider the element discussed in this first paragraph to comply with there main objectives and the state of their findings.

I do not understand how you can state “This highlights the sensitivity of NIRS to detect muscle oxygenation deficits following EIMD depends on both the exercise modality employed and the specific measurement and data processing techniques utilised.”. Your findings about the TSI during sprint indicate large effect by for only two variables, and do not reflect a clear impairment. Unless I missed some information, I suggest the author to rephrase this sentence to comply with true findings.

**Do you want your identity to be public for this peer review?** For information about this choice, including consent withdrawal, please see our Privacy Policy

Reviewer #1: No

Reviewer #2: No

---

## [Author Response · Author response to Decision Letter 1]

26 Nov 2025

Response to Reviewers doc has been added to the files submitted.

---

## [Decision Letter · Decision Letter 1]

22 Dec 2025

Dear Dr. McManus,

Thank you for submitting your manuscript to PLOS ONE. After careful consideration, we feel that it has merit but does not fully meet PLOS ONE’s publication criteria as it currently stands. Therefore, we invite you to submit a revised version of the manuscript that addresses the points raised during the review process.

We look forward to receiving your revised manuscript.

Kind regards,

Laurent Mourot

Section Editor

PLOS One

Journal Requirements:

Reviewers' comments:

Reviewer's Responses to Questions

**Comments to the Author**

Reviewer #1: All comments have been addressed

Reviewer #2: All comments have been addressed

2. Is the manuscript technically sound, and do the data support the conclusions?

Reviewer #1: Yes

Reviewer #2: Yes

3. Has the statistical analysis been performed appropriately and rigorously?

Reviewer #1: Yes

Reviewer #2: Yes

4. Have the authors made all data underlying the findings in their manuscript fully available?

Reviewer #1: Yes

Reviewer #2: Yes

5. Is the manuscript presented in an intelligible fashion and written in standard English?

Reviewer #1: Yes

Reviewer #2: Yes

Reviewer #1: The authors did a serious and meticulous job in revising their paper. I have not further comments at this stage

Reviewer #2: Authors provided a commendable revision of their manuscript that enhance its clarity. Considering the comments of the reviewer #1 and the associated corrections, I have some minor comments to address in order to fix the remaining concerns.

L 301: Considering the answer provided to reviewer #1, authors mentioned that power output was provided to characterize exercise performance level of this population. The use of the mean power as a reflect of the whole performance during sprint 1 remains however of modest information to translate exercise capacity and the presence of residual fatigue. Did the authors attempt to analyze data about total work provided during sprint (i.e. integral from the torque-time signal)? This element should be more suitable to detect the presence of residual fatigue and performance decrement than the mean value calculated throughout the sprint.

In the discussion section (no line reference available), there is a mention that a small but consistent upward shift in TSI was observed compared to baseline. This element seems to be determined by the small effect size reported in table 6, but not supported by statistics. Did the authors also consider the smallest worthwhile change for this parameter to support their assumption?

Typing error page 80: field compatable assessments : field compatible assessments?

The conclusion that “EIMD-related changes in muscle oxygenation are detectable with NIRS is likely to depend on both the exercise modality and the specific measurement and data processing techniques used” is not convincing, are rather surprising. Particularly, I am not convinced that “specific measurements and data processing” represent a valid hypotheses to present about physiological responses to eccentric exercise. In my opinion, authors should better rephrase this sentence to provide a clear explanations about whether or not EIMD could be detected and monitored following exercise.

**Do you want your identity to be public for this peer review?** For information about this choice, including consent withdrawal, please see our Privacy Policy

Reviewer #1: No

Reviewer #2: No

---

## [Author Response · Author response to Decision Letter 2]

8 Jan 2026

Response to Reviewers submitted as Word attachment

---

## [Decision Letter · Decision Letter 2]

15 Jan 2026

Assessment of Muscle Oxygenation Following Eccentric Exercise-Induced Muscle Damage Using Near-Infrared Spectroscopy

PONE-D-25-37045R2

Dear Dr. McManus,

We’re pleased to inform you that your manuscript has been judged scientifically suitable for publication and will be formally accepted for publication once it meets all outstanding technical requirements.

Kind regards,

Laurent Mourot

Section Editor

PLOS One

Additional Editor Comments (optional):

Reviewers' comments:

Reviewer's Responses to Questions

**Comments to the Author**

Reviewer #2: All comments have been addressed

2. Is the manuscript technically sound, and do the data support the conclusions?

Reviewer #2: Yes

3. Has the statistical analysis been performed appropriately and rigorously?

Reviewer #2: Yes

4. Have the authors made all data underlying the findings in their manuscript fully available?

Reviewer #2: Yes

5. Is the manuscript presented in an intelligible fashion and written in standard English?

Reviewer #2: Yes

Reviewer #2: (No Response)

**Do you want your identity to be public for this peer review?** For information about this choice, including consent withdrawal, please see our Privacy Policy

Reviewer #2: No

---

## [Editor Report · Acceptance letter]

PONE-D-25-37045R2

PLOS One

Dear Dr. McManus,

I'm pleased to inform you that your manuscript has been deemed suitable for publication in PLOS One. Congratulations! Your manuscript is now being handed over to our production team.

Kind regards,

on behalf of

Dr Laurent Mourot

Section Editor

PLOS One